# External Loads in Under-12 Players during Soccer-7, Soccer-8, and Soccer-11 Official Matches

**DOI:** 10.3390/ijerph18094581

**Published:** 2021-04-26

**Authors:** Mario Sanchez, Rodrigo Ramirez-Campillo, Daniel Hernandez, Manuel Carretero, Jesus Maria Luis-Pereira, Javier Sanchez-Sanchez

**Affiliations:** 1Research Group Planning and Assessment of Training and Athletic Performance, Universidad Pontificia de Salamanca, 37007 Salamanca, Spain; mcarreterogo@upsa.es (M.C.); jsanchezsa@upsa.es (J.S.-S.); 2Human Performance Laboratory, Department of Physical Activity Sciences, Universidad de Los Lagos, Osorno 5290000, Chile; r.ramirez@ulagos.cl; 3Centro de Investigación en Fisiología del Ejercicio, Facultad de Ciencias, Universidad Mayor, Santiago 7500000, Chile; 4Faculty of Education, Universidad Pontificia de Salamanca, 37007 Salamanca, Spain; danielhdez10@hotmail.com (D.H.); jesuspereira77@gmail.com (J.M.L.-P.)

**Keywords:** children, youth sports, fatigue, injuries, physical exercise

## Abstract

The aim of this study was to compare the external loads (i.e., displacement distances and velocities) of 10–11 years-old soccer players during Soccer-7 (i.e., seven-a-side), Soccer-8 (i.e., eight-a-side), and Soccer-11 (i.e., eleven-a-side) official matches. Male athletes (*n* = 133; age, 10.9 ± 0.8 years) were measured during official matches for total distance (TD), relative distance (Drel), maximal velocity (Vmax), acceleration (ACC), deceleration (DEC), and absolute and relative distance covered at different velocities. Data during matches were collected using a Global Positioning System unit. Greater TD was recorded during Soccer-11 compared to Soccer-7 and Soccer-8 (*p* < 0.01), and greater Drel during Soccer-11 compared to Soccer-8 (*p* < 0.05). Absolute ACC was greater during Soccer-11 compared to Soccer-7 (*p* < 0.01), although relative values for %ACC and %DEC were greater during Soccer-7 and Soccer-8 compared to Soccer-11 (*p* < 0.01). Globally, results show that Soccer-11 matches induce greater external loads compared to Soccer-7 and Soccer-8 matches. Current results may help coaches and soccer-related organizers to plan more suited soccer competitions for young players, with lower external loads.

## 1. Introduction

Performance markers have been used in sports science to identify, characterize, and potentiate training methodologies [1]. In soccer, the analysis of performance markers has been conducted during games by time-motion and notational analysis [2]. This analysis has indicated that soccer imposes high metabolic and physiological stress during competition, involving non-cyclical efforts over prolonged periods of time [3]. During a competitive match, soccer players may perform ~1400 short-duration maximal-intensity activities, including sprints, change of directions, tackling, accelerations, decelerations, jumps, among others [4]. Moreover, soccer players may cover up to 13-km during a match [5]. 

Match analysis research has extensively studied senior male players of sub-elite to elite competitive standard [6]. Senior football matches are played on pitches sized 100–110 × 64–75 m, whereas pitch size varies especially in youth football matches, with the pitch size and number of players adapted to different age groups [7]. Although there is information available regarding different formats played by soccer players of 12–17 years [8,9,10], research on younger players (i.e., under-12) is scarce [11,12]. Studies carried out with this age group have used friendly matches [13] and training activities, such as small-sided games [14].

While some information is available about the physical and physiological demands of highly trained young soccer players during match play, little is known about whether match-related configurations (i.e., soccer-7, soccer-8, and soccer-11) affect external loads. In one study [15], no differences in walking, running, inactivity, and jumping were observed between Soccer-11 versus Soccer-7 matches in under-12 players. Another study that analyzed the total distance covered during Soccer-5, Soccer-8, and Soccer-11 matches in under-10 and under-13 players found no differences in under-10 player (i.e., Soccer-5 vs. Soccer-8) and values greater during Soccer-11 in under-13 players [7]. Similarly, greater external loads were observed during Soccer-11 versus Soccer-7 among under-12 players, as well as in under-14 and under-16 players [16]. Finally, a study that analyzed the activity profile of girls soccer players under-11 during Soccer-7 and Soccer-8, found no differences in the total distance covered and the speed peak, while the distance at high speed (>16 km∙h^−1^) was higher in Soccer-8 than in Soccer-7 [17]. The different results may be due to the fact that previous studies used multiple dimensions of the playing space.

To assess the external loads during competition (match analysis) a common technique is the use of Global Positioning System (GPS) units [18]. The positioning systems were used for military or scientific purposes, however, their development has given rise to wider applications, for example, making them available for use in team ports in order to quantify external training load [19]. The GPS units represent a comfortable option given its relatively low cost, low weight and size, easy data-collection and analysis process, and the ability to collect relevant data such as maximal velocity, accelerations, decelerations, distance covered, and distance covered at different velocities [20]. 

There is little published information about youth soccer, for which the dimensions of the playing surface and the number of team members may vary according to national guidelines [15]. Using the optimal competition model in youth soccer is essential to have an effective training process. This study, based on the analysis of the physical demand of different match formats, aims to provide information on which format is better for under-12 players. Therefore, the aim of this study was to compare the external loads (i.e., displacement distances and velocities) of under-12 soccer players during Soccer-7, Soccer-8, and Soccer-11 official matches. We hypothesized that the external loads during Soccer-11 would be greater compared to those induced by Soccer-7 and Soccer-8 [16,17].

## 2. Materials and Methods

### 2.1. Participants

Male under-12 soccer players (*n* = 133; age, 10.9 ± 0.8 years; height, 145.4 ± 5.5 cm; body mass, 37.1 ± 4.1 kg) participated in this study. Soccer players participated in Soccer-7 matches (*n* = 49), Soccer-8 matches (*n* = 46), and Soccer-11 matches (*n* = 38). All teams participated in provincial leagues of the category and had played the territorial sector in the previous season. The total number of records used for the analysis was 5 in Soccer-7, 5 in Soccer-8 and 4 in Soccer-11. All soccer players regularly trained 90-min three times per week without athletic physical conditioning, or strength or power training. The training sessions were: (i) First session, for the development of physical condition through small-sided games tasks, (ii) second session, aimed at technical improvement, (iii) third session, dedicated to improving collective tactical behaviors. In addition to the training, the teams competed one time every weekend. The participants were soccer players from Spanish Professional League Teams Academies. All players were trained by coaches with UEFA “A” license. To be included in the study, soccer players needed (i) to be free from injuries in the four months before the study and (ii) to be involved in soccer training and competition for at least four years. Goalkeepers were not included in the analysis. The technical staff of the soccer club gave full support to conduct the study. Participants, as well as their respective parents or legal guardians, were informed about the experimental procedures and their possible risks and benefits before the start of the study. Written informed consent of legal representatives was obtained before the beginning of the study. Soccer players were free to withdraw from the study without giving any reasons and without any penalty regarding their academy position. This study protocol was in accordance with the latest version of the Declaration of Helsinki.

### 2.2. Procedures

Before the match, stature was measured using a stadiometer (Seca 214 Road Rod Portable; Seca^®^, Ltd., Hanover, Germany, to 0.5 cm) and body mass on an electrical scale (Tanita BC-418 MA Segmental; Tanita^®^, Tokyo, Japan, to 0.1 kg). Before the warm-up of each game [21], a GPS unit (K-Gps 10 Hz; K-Sport^®^, Montelabbate, PU, Italy) was used to quantify activity profiles during official matches as previously validated in soccer players [22]. The GPS unit was fixed to the torso of each player, inserting it in the vest that the players wore under the official competitive uniform t-shirt. At the end of each game, the GPS unit was turned off. A researcher recorded the periods in which the player was not in play: (i) Warm-up and down time, (ii) rest between first and second period, (iii) reserve players’ bench. The GPS-collected data (distance, speed, acceleration, and deceleration) were downloaded using the software KFitness (K-Sport^®^, Montelabbate, PU, Italy), identifying total game time of each player, without considering warm-up time, substitution time (i.e., soccer-8 and soccer-7), or rest time between match halves.

Fourteen official matches were analyzed, being five from Soccer-7, five from Soccer-8, and four from Soccer-11. Each match had two halves of 30-min, with an inter-halve rest of 10-min. Athletes played each match after ≥48 h of rest from their last training session or competition. During the game, the players consumed ad libitum water. All participants played as the local team. Given that players completed different minutes of play during Soccer-7, Soccer-8, and Soccer-11, aside from absolute values of external loads (distances and velocities), values were divided by played time (i.e., relative values).

According to game rules, the Soccer-7 and Soccer-8 matches were performed on 40 × 55 m pitches (173 m^2^ and 138 m^2^ per player, respectively), whereas Soccer-11 matches were performed on 68 × 105 m pitches (325 m^2^ per player). All games were played on artificial grass. During the Soccer-7 and Soccer-8 matches, unlimited substitutions were allowed, with the possibility for a player to return to play after the substitution, whereas a maximum of five substitutions were allowed during Soccer-11 matches, without the possibility to return to play after the substitution. Moreover, given that rules for Soccer-7 and Soccer-8 indicate that all players must complete ≥20 min during a match, for the current study, data from players were included only if the athlete completed ≥20 min of play during Soccer-7, Soccer-8, and Soccer-11 competitions [23].

According to previous recommendations [23], match data analyzed were: Total distance (DT; m), peak velocity (Vmax; km∙h^−1^), DT relative to match duration (Drel; m∙min^−1^), total distance completed in acceleration (ACC; 1.5 m∙s^−2^), total distance completed in deceleration (DEC; −1.5 m∙s^−2^), percentage of DT completed in acceleration (%ACC = [DT/ACC] × 100), percentage of DT in deceleration (%DEC = [DT/DEC] × 100).

In addition, covered distances at different velocities were also analyzed: 0–0.4 km∙h^−1^ (DV1; standing), 0.5–3.0 km∙h^−1^ (DV2; walking), 3.1–8.0 km∙h^−1^ (DV3; jogging), 8.1–13.0 km∙h^−1^ (DV4, medium-intensity running), 13.1–18.0 km∙h^−1^ (DV5, high-intensity running), ≥18.1 km∙h^−1^ (DV6; sprinting). Moreover, covered distances at different velocities were also expressed in relation to DT as follows: %DV1 = [DT/DV1] × 100); %DV2 = [DT/DV2] × 100); %DV3 = [DT/DV3] × 100); %DV4 = [DT/DV4] × 100); %DV5 = [DT/DV5] × 100); %DV6 = [DT/DV6] × 100).

### 2.3. Statistical Analyses

Data are presented as mean ± SD. Data normality was checked with the Shapiro–Wilk test. To compare dependent variables between matches, one-way ANOVA test was used with Tukey HSD post hoc and α at *p* ≤ 0.05. In addition, Cohen’s d effect size (ES) with 90% confidence limits was used. Ranges for ES analysis were set at <0.1 (very small), 0.1–0.2 (small), 0.2–0.5 (moderate), 0.5–0.8 (high), y > 0.8 (very high) [24]. Data analysis was completed using Statistical Package for Social Sciences (SPSS, v. 21.0; SPSS, Inc., Chicago, IL, USA).

## 3. Results

Greater (*p* < 0.01) DT was observed during Soccer-11 (4598 ± 1163 m) compared to Soccer-7 (3064 ± 822 m) and Soccer-8 (3364 ± 1229 m). In addition, greater Drel was observed in Soccer-11 (91.27 ± 9.56 m∙min^−1^) compared to Soccer-8 (84.88 ± 7.37 m∙min^−1^; *p* < 0.05). No differences (*p* > 0.05) in Drel were observed between Soccer-8 and Soccer-7 (Figure 1). 

Table 1 shows absolute (m) and relative (%) distances covered at different velocities. Greater (*p* < 0.01) absolute DV2 was observed in Soccer-11 than Soccer-7. Moreover, greater (*p* < 0.05) absolute DV1, DV3, DV4, DV5, and DV6 was observed in Soccer-11 compared to Soccer-7 and Soccer-8. Lower (*p* < 0.05) %DV1 and %DV2 was observed in Soccer-11 than Soccer-8. Moreover, lower (*p* < 0.05) % DV5 and %DV6 was observed in Soccer-8 than Soccer-7.

Figure 2 depicts absolute and relative ACC and DEC values. Greater (*p* < 0.01) ACC was observed in Soccer-11 (390 ± 116 m) compared to Soccer-7 (307 ± 88.7 m). However, in relative terms, greater (*p* < 0.01) %ACC (Soccer-7 = 10.0 ± 1.0%; Soccer-8 = 10.5 ± 1.3%; Soccer-11 = 8.4 ± 1.5%) and %DEC (Soccer-7 = 10.3 ± 1.0%; Soccer-8 = 9.8 ± 1.4%; Soccer-11 = 7.9 ± 1.5%) were observed in Soccer-7 and Soccer-8 compared to Soccer-11.

Similar Vmax was observed in Soccer-7 (23.2 ± 1.6 km·h^−1^), Soccer-8 (22.7 ± 1.7 km·h^−1^), and Soccer-11 (23.2 ± 1.7 km·h^−1^).

## 4. Discussion

The aim of this study was to compare the external loads (i.e., displacement distances and velocities) absolute and relative, of under-12 soccer players during Soccer-7, Soccer-8, and Soccer-11 official matches. The main results revealed greater absolute DT, Drel, DV1, DV2, DV3, DV4, DV5, DV6, and ACC during Soccer-11 compared to Soccer-7 and Soccer-8. The %DV1, %DV2, %ACC, and %DEC were lower during Soccer-11 compared to Soccer-7 or Soccer-8. Overall, these findings revealed greater external loads during Soccer-11 in under-12 male soccer players compared to Soccer-7 and Soccer-8 matches. In Soccer-7 and Soccer-8, the area per player is smaller (i.e., 57.5% and 46.8%, respectively) than in Soccer-11. In general, greater pitch dimensions impose a greater physical load on the player [25], which also seems applicable to under-12 players.

Greater DT was observed in Soccer-11 than Soccer-7 and Soccer-8. These results are in line with previous studies that observed greater DT in under-12 players during Soccer-11 than Soccer-7 [16], and no difference between Soccer-7 and Soccer-8 in under-11 female soccer players [17]. The greater DT observed in Soccer-11 than other modalities (i.e., Soccer-7 and Soccer-8) may be related to the number of players on the team, the pitch dimensions [26,27], and technical-tactical aspects [11]. In this sense, the regulations in Soccer-11 force under-12 players to be more separated to occupy the playing space [28], which increases the physical load to maintain certain collective behaviors [29]. Therefore, current results indicate that each modality is associated with a DT and if trainers can know the running distances during matches they will be able to prepare training programs that are specific to the players’ age and ability [30]. Using the ideal training load on each player is very important since there is a close link between training intensity and positive outcomes in relation to cardiovascular, metabolic, and musculoskeletal health [17]. For this reason, Soccer-11 would be appropriate for under-12 players with better physical condition and optimal technical-tactical performance to respond effectively to the demands of the game [31].

Although DT is commonly used to assess the external load for match analysis purposes, when players complete different total playing time, the Drel may best reflect the external load imposed on a given player [2]. This marker is especially important considering that match rules for Soccer-7, Soccer-8, and Soccer-11 may induce different playing times among players [10]. In the current study, a greater Drel in Soccer-11 was observed compared to Soccer-8. Although in our work, we did not find differences between Soccer-11 and Soccer-7, other studies indicated that the Drel was greater in Soccer-11 [16]. Drel depends on competitive standard, tier, playing position, phase of the season, and physical fitness [6,32]. Although physical fitness was not controlled in this study, the level of endurance fitness has been used to explain differences in external load in soccer matches [12]. With respect to playing position, this variable was not analyzed since coaches of youth soccer frequently interchanged players during matches to improve the technical-tactical abilities [11].

The distance covered in DV1, DV3, DV4, DV5, and DV6 was greater in Soccer-11 than in Soccer-7 and Soccer-8. These findings are similar to those reported by Mora et al. [16], showing greater covered distance at ≥13 km·h^−1^ in Soccer-11 compared to Soccer-7. Other studies with under-13 players also found more distance to sprint in Soccer-11 than Soccer-8 [7]. In the present study, the area per player was considerably greater in Soccer-11 (i.e., 325 m^2^) than Soccer-7 and Soccer-8 (173 m^2^ and 138 m^2^, respectively), which could be one reason for the higher distance high speed [25]. Moreover, a greater distance between goals in Soccer-11 can allow the player to run at high speed [17]. Although the distance covered at high speed is an indicator of the physical demand associated with each type of game [6], these results should be taken with caution. On the one hand, in the previous studies, different speed thresholds that were used to define high-intensity may have altered the results [11]. Moreover, arbitrary thresholds were used in most studies, but to improve the comparison between players, age-specific speed thresholds should be created based on the peak velocity of a 10-m sprint [10]. 

In contrast to previous studies [16], ACC was greater in Soccer-11 than in Soccer-7. These results are different from those described in the literature, where it is indicated that a smaller area per player brings closer proximity in the participants, increasing the frequency of acceleration, deceleration, and change of direction actions [15,24]. The analysis of the relative values indicated higher values in %ACC and %DEC in Soccer-7 and Soccer-8 than in Soccer-11. This may suggest that there is a development in the neuromuscular ability during a Soccer-7 and Soccer-8, adapted to the player’s possibilities. In addition, due to the possibility of unlimited changes in Soccer-7 and Soccer-8, the Soccer-11 players did have a significantly longer playing time, which may have affected the number of ACC performed in this match format. In Soccer-11, it may be advisable to introduce rules to control this greater neuromuscular load that is related to eccentric actions, which can affect fatigue and recovery processes post-match and the risk of injury in soccer players [33].

Finally, regarding the Vmax performed by under-12 players, as in other studies [16], no differences were observed between game modalities. Although previous studies indicated that the pitch size influences the possibility of reaching the highest running speeds [25], it appears that Vmax was affected by training status [34]. A large effect on peak speed was observed, with elite players having a 2.8 km·h^−1^ higher peak speed than recreational players [7]. Although all participating teams were included in the elite category under-12, was not taking into account the degree of biological maturation of the players. This is especially critical considering that under-12 soccer players may exhibit a limited capacity to repeat short-term maximal-intensity actions [35].

Understanding the match-play demands of elite youth soccer could have practical implications to consider the best competition format to include in the soccer teaching process. The training in Soccer-7 and Soccer-8 can be based on small-sided game formats because they are less complex due to fewer interactions between teammates and opponents, thus making the game more individualized, in which many actions of high neuromuscular involvement are developed. Coaches must manage these tasks with caution, as they are very intensive for the player. Soccer-11 players must train in medium and large matches, to develop speed actions and increase distance covered. Coaches should schedule the task so that these actions are not repeated and fatigue can be avoided. Moreover, understanding the match-play demands of elite youth soccer could have practical implications to consider the best competition format to include in the soccer teaching process.

A potential limitation to the study is that not all players completed the game formats. However, this allowed increasing the ecological validity of the study. On the other hand, certain individual aspects of the players could have explained some results. Finally, only one game was analyzed per player in each format. Game-to-game variability has been shown to be high, especially in high-speed running zones for adults [36], and it is likely that a similar pattern would be observed for youth players.

## 5. Conclusions

Soccer-11 matches imply a higher physical load for under-12 players than other modalities, such as Soccer-7 and Soccer-8. In Soccer-11, the players cover greater total distance and distance at high speed, and they also have to a greater neuromuscular support.

## Figures and Tables

**Figure 1 ijerph-18-04581-f001:**
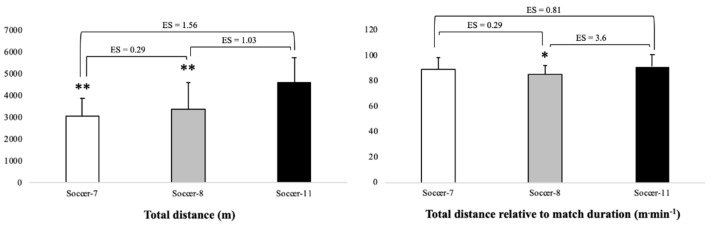
Total distance and total distance relative to match duration during Soccer-7, Soccer-8, and Soccer-11 official matches. ES: effect size; * and **: Significant differences versus Soccer-11 (*p* < 0.05 and *p* < 0.01, respectively).

**Figure 2 ijerph-18-04581-f002:**
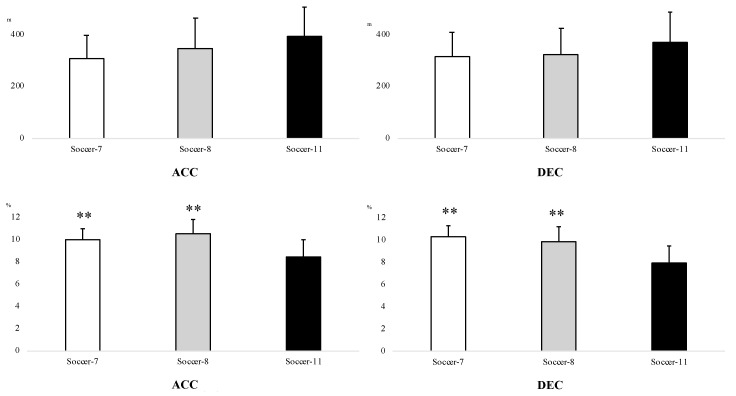
Absolute (m) and relative (%) distance covered in acceleration (ACC) and deceleration (DEC) during Soccer-7, Soccer-8, and Soccer-11 official matches among under-12 players. ES: Effect size. **: Significant difference versus Soccer-11 (*p* < 0.01).

**Table 1 ijerph-18-04581-t001:** Distance covered at different speeds during Soccer-7, Soccer-8, and Soccer-11 official matches.

Measures	Modalities	DV1 (Standing)	DV2 (Walking)	DV3 (Jogging)	DV4 (Medium-Intensity Running)	DV5 (High-Intensity Running)	DV6 (Sprinting)
m	Soccer-7	12.0 ± 3.4 **	238 ± 102 **	1054 ± 312 **	1136 ± 355 **	503 ± 170 **	120 ± 69.8 *
	Soccer-8	13.9 ± 5.2 **	292 ± 118	1189 ± 453 **	1305 ± 516 **	473 ± 239 **	92.1 ± 69.4 **
	Soccer-11	16.7 ± 4.9	341 ± 129	1590 ± 438	1773 ± 552	705 ± 253	171.4 ± 133
%	Soccer-7	0.4 ± 0.1	7.8 ± 2.7	34.5 ± 4.5	36.7 ± 3.7	16.5 ± 3.7 ^##^	4.0 ± 2.1 ^#^
	Soccer-8	0.4 ± 0.1 **	8.9 ± 2.3 *	35.4 ± 4.6	38.6 ± 3.9	14.0 ± 4.2	2.7 ± 1.8
	Soccer-11	0.3 ± 0.1	7.6 ± 2.6	34.6 ± 5.0	38.9 ± 6.1	15.3 ± 4.0	3.8 ± 2.6
ES m	Soccer-7 vs. Soccer-11	1.13	0.91	1.44	1.41	0.96	0.51
	Soccer-7 vs. Soccer-8	0.43	0.49	0.35	0.38	0.14	0.41
	Soccer-8 vs. Soccer-11	0.55	0.40	0.91	0.88	0.95	0.77
ES %	Soccer-7 vs. Soccer-11	0.42	0.11	0.02	0.34	0.31	0.09
	Soccer-7 vs. Soccer-8	0.34	0.42	0.19	0.49	0.65	0.65
	Soccer-8 vs. Soccer-11	0.74	0.55	0.17	0.03	0.33	0.48

DV1, DV2, DV3, DV4, DV5, and DV6: Distance covered at velocities of 0–0.4 km∙h^−1^, 0.5–3.0 km∙h^−1^, 3.1–8.0 km∙h^−1^, 8.1–13.0 km∙h^−1^, 13.1–18.0 km∙h^−1^, ≥18.1 km∙h^−1^; * and **: significant differences versus Soccer-11 (*p* < 0.05 and *p* < 0.01, respectively); ^#^ and ^##^: significant differences versus Soccer-8 (*p* < 0.05 and *p* < 0.01, respectively).

## Data Availability

Not applicable.

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
