# Peer review of "External Loads in Under-12 Players during Soccer-7, Soccer-8, and Soccer-11 Official Matches"

_ijerph, 2021, doi:10.3390/ijerph18094581_

Round 1

Reviewer 1 Report

The objective of the study was to compare the physical demand of different types of competition in young soccer players. The manuscript can provide information to the training process in youth soccer players. In addition, it can help to select the best type of competition, within the soccer learning process. In this sense, the results seem to recommend a competition with few players, a small playing space and free changes during match time. Although the study design is correct, the manuscript is written with precision and clarity, it is necessary for the authors to take into account some issues to improve the manuscript.

Introduction

- During the text you should not use U12 (and other ages), they should refer to under-12. They should apply it to all text

- Ln45. The authors must specify that the information available to date is related to friendly matches and reduced games.

- Ln 58. Why are there such different results between the studies?

- Ln 63. The information from the GPS units must be more accurate. In addition, references on its practical application and measurement systems can be included.

Materials and Methods

-Ln 76. Remove "absolute and relative distance covered" by "distance covered ..."

- Ln82. The total registration number per competition must be included.

- Ln 83. The type of practice performed during the training time should be included.

- Ln 86. A space must be removed after the period.

- Ln 85. The level of competition must be indicated.

- Ln 104. When was the player not in play? This idea should be better expressed.

- Ln109. Remove Analyzed by Analyzed

- Ln 109. How many matches correspond to each type of competition?

- Ln 116. Remove “According to Soccer-7, Soccer-8 and Soccer-11 game rules, the Soccer-7 and Soccer-8…” by “According to game rules, the Soccer-7 and Soccer-8…”

Results

-Ln 150. Remove “m/min” by “m∙min-1

-Ln 152. The last sentence of the paragraph can be deleted

-Ln 160. The speed ranges are expressed in values relative to the total distance. Therefore, the variable is not DV1, it is% DV1. Check in all

-Table 1. Check spaces in column text

Discussion

- During the text you should not use U12 (and other ages), they should refer to under-12. They should apply it to all text

-Ln 184. How was that percentage calculated?

-Ln187. Remove “This results” by “These results”

- Ln 199. The authors should indicate a practical application to these results.

-Ln 209. Was the fitness level of the players analyzed during the study?

- Ln 212. The paragraph must end with an application phrase for this variable

- Ln 214. The reference to Table 1 does not need to appear in this section

-Ln 233. Paragraphs must be joined

Physical performance during soccer-7 competition and small-sided games in U12 players

J Sanchez-Sanchez, M Sanchez, D Hernández, O Gonzalo-Skok, ...

Journal of Human Kinetics

Influence of different small-sided games formats on physical and physiological demands and physical performance in young soccer players

D Castillo, A Rodríguez-Fernández, FY Nakamura, J Sanchez-Sanchez, ...

Journal of strength and conditioning research

Effects of short-term in-season break detraining on repeated-sprint ability and intermittent endurance according to initial performance of soccer player

A Rodríguez-Fernández, J Sanchez-Sanchez, R Ramirez-Campillo, ...

PLoS ONE 13 (8), e0201111

Reviewer 2 Report

The current study investigated whether there are differences in performance markers due to the soccer match settings. While it is a well-executed study, the results are fairly intuitive (i.e., Soccer-11 matches induce greater external loads compared to Soccer-7 and Soccer-8) due to the pitch size, game rules (e.g., player substitutions), and game tactics. Firstly, you need to come up with a better justification why this study should be conducted and why we need to know more about the relationship between match-related configurations and performance markers/external loads (as is the case with elite-level soccer, the fact that Soccer-11 induce greater loads is intuitive and expected). Secondly, the authors may consider different analytical strategies (e.g., ANCOVA) to player-related information. Also, it is advised to provide demographic information of the research participants per group to see how player-level variables might have influenced the study results. Thirdly, in the discussion section, you did well in explaining the results but did not provide 'practical' implications and coaching/training-related recommendations. Fourthly, in Figure 1 and Table 1, some ES scores were greater than 1.0. Lastly, I think it will be much more meaningful to discuss more the differences between Soccer 7 and Soccer 8 because they use the same pitch size. In addition, Line 143; it should be "0.2 - 0.5 (moderate)", not 1.5.

Hope this helps. 

Round 2

Reviewer 1 Report

The manuscript has improved adequately and from my point of view, it would be recommended for publication

Author Response

Thank you for you review.

Best regards.

Reviewer 2 Report

You still need to discuss more the practical implications - maybe you can use some examples to illustrate what differences coaches should make per match styles in terms of training methods. 

Author Response

Thanks for your comment. The requested section has been discussed further, providing examples that will help coaches improve their performance in daily practice (training sessions). The new text can be found between lines 259 to 269.